# The Problem of Weed Infestation of Agricultural Plantations vs. the Assumptions of the European Biodiversity Strategy

Adrianna Kubiak [1], Agnieszka Wolna-Maruwka [1,*], Alicja Niewiadomska [1] and Agnieszka A. Pilarska [2]

[1] Department of Soil Science and Microbiology, Poznan University of Life Sciences, Szydłowska 50, 60-656 Poznan, Poland; adrianna_kubiak@interia.pl (A.K.); alicja.niewiadomska@up.poznan.pl (A.N.)
[2] Department of Hydraulic and Sanitary Engineering, Poznan University of Life Sciences, Piątkowska 94A, 60-649 Poznan, Poland; pilarska@up.poznan.pl
* Correspondence: amaruwka@up.poznan.pl; Tel.:+48-618-466-724

**Abstract:** Meeting the nutritional needs of a dynamically developing global society is a major challenge. Despite the modernisation of agriculture, huge losses in the quality and quantity of crops occur each year, mainly due to weed species, which are the most important biotic limitation to agricultural production. Globally, approximately 1800 weed species cause a 31.5% reduction in plant production, which translates to USD 32 billion per year in economic losses. However, when the same herbicides are frequently applied, plants develop segetal immune mechanisms. There are currently around 380 herbicide-resistant weed biotypes worldwide. Due to the negative influence of herbicides on ecosystems and the legal regulations that limit the use of chemical crop protection products, it is necessary to develop a new method of weed control. Bioherbicides, based on living organisms or their secondary metabolites, seem to be an ideal solution. The biocontrol market is worth around EUR 550 million in Europe and EUR 1.6 billion worldwide, with an estimated 15% growth expected by 2025. Despite numerous studies that have demonstrated the effectiveness of microbial bioherbicides, only 25 mould-based bioherbicides are currently available to growers. Due to the high specificity and selectivity of biological crop protection products, as well as their low production costs and non-toxicity to the environment and human health, they would appear to be a safe alternative to chemical pesticides.

**Keywords:** weeds; pesticides; bioherbicides; microorganisms; sustainable farming

## 1. Introduction

Weed plant species have a significant negative impact on the productivity of agricultural crops, and their control is a major challenge for the agricultural sector [1]. These plants can rapidly capture limited natural resources, such as water, light, soil nutrients and space. They can reproduce faster than cultivated plants because of features such as a deep root system, resistance to drought and frost, and high nutrient use efficiency. Moreover, weeds can release allelopathic substances into the soil, and support the development of pests and crop pathogens. These properties make them competitive with arable crops, often leading to a reduction in crop yield and, at the same time, an increase in cultivation costs [2].

Unfortunately, excessive and often inappropriate use of chemical herbicides has resulted in a number of serious side effects, which include weed resistance to these substances, and soil and groundwater contamination, as well as harmful effects on non-target organisms. Although they are economically profitable crop protection products, the public's concerns about their harmful effects on the environment are growing. Weed control based on the use of biological methods is an alternative to chemical-based products as the consequences of the negative impact of herbicides on the environment, and, hence, human and animal health, are currently of seriousconcern [3,4]. Bailey [5] defines bioherbicides as "products of natural origin intended for weed control, which may contain living organisms, more specifically microorganisms or their natural metabolites".

Under the European Green Deal [6–8] and Integrated Pest Management [9,10], it is a challenge for modern agriculture to maintain the current level of crop production, while reducing pesticide use by up to 50%. In recent years, the price of crop protection products has increased by 15–25%. This is an additional problem for European agriculture as farmers are forced to search for alternative methods to continue to obtain optimal crop yields. The assumption of integrated pest management and the European Biodiversity Strategy is to reduce the use of chemical crop protection products in order to protect the environment and to reverse the degradation of ecosystems.

The fundamental goals of the Food and Agriculture Organization (FAO) of the United Nations are (i) the development of sustainable agriculture, thereby ensuring that all people have equal access to high-quality food, and (ii) global food security [11].

The European Biodiversity Strategy includes proposals for actions that will have a significant impact on the agricultural sector in the European Union (EU). They include restrictions that not only affect the use of plant protection products, fertilisers and antimicrobials, but also stimulate the development of the organic farming sector, change the eating habits of European consumers, protect and restore ecosystems, and increase the biodiversity of natural resources.

In facing the assumptions outlined in the European Biodiversity Strategy, there is a lack of research addressing the fundamental problem of weed infestation on farmlands in Europe on the one hand, and the issue of environmentally safe methods of combating those weed species (with both chemical and biological methods) on the other. In this paper, we review the international literature and present the research that has been carried out to date on the above-mentioned issues.

The objective of this review is to highlight the problem of weed infestation on farmlands in Europe, and identify the environmental threats that result from the use of herbicides in the context of the European Biodiversity Strategy. This conceptual review also highlights the practicalities of using bioherbicides, which a farmer can employ to reduce or completely eliminate the use of chemical plant protection products on their farm. The most important aspects of this assumption are: (i) the proven effectiveness of bioherbicides; (ii) a high degree of selectivity and specificity; and (iii) non-toxicity to the environment, humans and animals.

## 2. Weed Infestation of Agricultural Plantations in Europe

### 2.1. Main Weed Species on Plantations of the Most Economically Important Crops in Europe

According to FAO reports [12], wheat (*Triticum aestivum* L.), sugar beet (*Beta vulgaris* L. subsp. *vulgaris*) and maize (*Zea mays* L.) were the main agricultural crops in Europe between 2018 and 2020.

Wheat is one of the most important cereals worldwide. It is grown in almost all countries and is the main food source for billions of people. Therefore, this crop is of fundamental importance in ensuring global food security [13]. In 2020, the global production of wheat amounted to about 760.9 million tonnes (Asia—45.7%, Europe—33.5%, Americas—15.5%, Africa—3.3%, and Oceania—2%). In 2020, the largest wheat producers in Europe were Russia (85.9 million tonnes), France (30.1 million tonnes), and Ukraine (24.9 million tonnes) [12]. As the world population is constantly growing, it is crucial to increase the production of wheat, which is a common food source. Segetal weeds are responsible for the greatest losses in wheat production efficiency. The study by Gaba et al. [14], carried out in western France on 150 experimental fields with winter wheat, showed the presence of 108 species of weeds (an average of 9.46 species per plot), dominated by *Fallopia convolvulus* L. A. Love, *Mercurialis annua* L., *Polygonum aviculare* L., *Veronica persica* Poir., *Galium aparine* L., and *Poa* sp. Hofmeijeret al. [15] observed 197 weed species in five European countries (Denmark, Finland, Germany, Latvia, and Sweden). They noted that the most common segetal plants in 207 spring cereal fields were *Stellaria media* L. Cirillo (relative frequency: 0.90), *Viola arvensis* Murray (relative frequency: 0.86), *Cirsium arvense* L. Scop. (relative frequency: 0.85), *Tripleurospermum inodorum* L. Sch. Bip. (relative frequency:

0.84) and *Chenopodium album* L. (relative frequency: 0.78). Work by Pinke et al. [16] in cereal fields in western Hungary (Lesser Plain of north-western Hungary, the Transdanubian Mountain range, the West Hungarian margin territory and the hilly region of southern Transdanubia) recorded the following weed species: *V. triphyllos* L., *Anthemis ruthenica* M. Bieb., *Myosotis stricta* Link ex Roem. and Schult., *Cerastium glomeratum* Thuill., *Vicia villosa* Roth., *Arabidopsis thaliana* L. Heynh., *Scleranthus annuus* L., *A. arvensis* L., *Apera spica-venti* L. P. Beauv and *Aphanes arvensis* L. Research by Cimalova and Lososova [17], in cereal fields in north-eastern Czech Republic (Northern and Central Moravia) documented the following weeds species: *A. arvensis*, *Matricaria discoidea* DC., *Papaver rhoeas* L., *Artemisia vulgaris* L., *Hypericum perforatum* L. and *Tanacetum vulga* L.

Poland is one of the leading producers of wheat in the EU, with 12.4 million tonnes of wheat produced in 2020 [12]. Harasim et al. [18] studied the effect of growth retardants (with various active ingredients) on the diversity of segetal plants on winter wheat, tested in eastern Poland on 10 m$^2$ experimental plots (in three replications). The researchers observed that the dominant weed species (total number: 96.3 pieces/m$^2$) in 2005–2007 in the control plots (without growth retardant) were: *V. persica*, *A. spica-venti*, *Viola arvensis* Murr., *V. arvensis* L., *Echinochloa crus-galli* L. P. Beauv, *C. album*, *Elymusrepens* L. Gould, *S. media*, and *Capsellabursa-pastoris* L. Medik. Sawicka et al. [19] identified 11 segetal weed species on winter wheat plantations in five commercial farms in south-eastern Poland. The most common species recorded were dicotyledonous weed species, such as *A. arvensis.*, *C. arvense*, *V. arvensis*, and *Equisetum arvense* L., with *P. convolvulus* L. and *Convolvulus arvensis* L. less common. The same researchers identified 14 segetal species on spring wheat plantations, where the proportionof dicotyledons *Brassica napus* L., *Sinapis arvensis* L., and *C. Arvense* amounted to 78.6% of all weed species.

Weed infestation also causes losses on sugar beet plantations worldwide. Due to the slow growth and wide spacing of sugar beet plants, these plants are very susceptible to weed infestation. Annual broadleaf weeds are a particular threat to sugar beet, because they grow taller than the crops and, thus, limit their access to sunlight. It is estimated that the uncontrolled growth and development of these species within the first eight weeks after the sowing of sugar beet or within four weeks after the crops have developed two leaves may reduce the yield by 26–100% [20]. According to FAO statistics [12], the global production of sugar beet in 2020 amounted to about 252.97 million tonnes (Europe—62.1%, Asia—18.3%, Americas—13%, and Africa—6.6%). The largest European sugar beet producers were Russia (33.9 million tonnes), Germany (28.6 million tonnes) and France (26.2 million tonnes). Bezhin et al. [21] compared the effectiveness of conventional and glyphosate-based weed control methods applied on sugar beet plantations in Russia and Germany between 2012 and 2014. Their research confirmed the hypothesis that weed infestation is related to the region; weed density was much greater in Russia and amounted to 82–237 weeds/m$^2$, while it was 28–61 weeds/m$^2$ in Germany. The following weed species were found in the plots: *E. crus-galli*, *Amaranthus retroflexus* L., *C. album*, *P. convolvulus* L. A. Love, *Lamium purpureum* L., *V. sativa* L., *Galeopsis tetrahit* L., *P. lapathifolium* L. subsp. *lapathifolium*, *C. arvensis* L., *P. aviculare*, and *V. arvensis*. Gerhards et al. [22] attempted to predict future sugar beet yield losses based on weed density, which ranged from 20 to 131 weeds/m$^2$ depending on the region. Aside from the plants identified by Bezhin et al. [21], the researchers also found *Fumaria officinalis* L., *Thlaspi arvense* L., *M. inodora* L., and *Setaria glauca* L. P. Beauv. Both Bezhin et al. [21] and Gerhards et al. [22] noted that the dominant weeds were *E. crus-galli* and *A. retroflexus*. According to Heno et al. [23], sugar beet plantations in France were most frequently infested by weeds such as *C. album*, *A. retroflexus*, *B. napus*, *M. annua* L., *Ammi majus* L., *C. arvense*, *V. arvensis*, *Sonchus arvensis* L., *F. officinalis*, *P. convolvulus* L., *P. aviculare*, *Alopecurus myosuroides* Huds., *Lolium perenne* L., and *Poa annua* L. These researchers also emphasised that *C. album* was the most important segetal species found on sugar beet plantations, regardless of the region.

For many years, sugar beet has also been one of the main economically important crops in Poland. In 2020, annual sugar beet production amounted to about 14.2 million

tonnes [12]. Domaradzki et al. [24] compared the weed infestation of sugar beet plantations in southern Poland between 1989 and 1995 and again between 2006 and 2012. During the first research period, they observed the presence of 36 segetal plant species, among which *C. album*, *A. retroflexus* L., *G. aparine*, *E. crus-galli*, *E. repens*. Gould, *M. maritima* L. ssp. *inodora* L. Dostal, *P. persicaria* L., *B. napus*, and *T. arvense* were dominant. During the second period, the researchers recorded 40 species of weeds. There was wide occurrence of the following species: *S. viridis* L. P. Beauv (cover index: 440), *C. album* (cover index: 3449), *B. napus* (cover index: 366), *P. lapathifolium* L. subsp. *Lapathifolium* (cover index: 705), *P. persicaria* (cover index: 858), *Galinsoga parviflora* Cav. (cover index: 368), *F. convolvulus* (cover index: 250), *C. arvensis* (cover index: 179), *G. aparine* (cover index: 166), *V. arvensis* (cover index: 154), *S. pumila* (cover index: 144), *E. crus-galli* (cover index: 141), *T. arvense* (cover index: 138), *A. retroflexus* (cover index: 112) and *M. maritima* (cover index: 102). In addition, the researchers identified ten new segetal weed species, i.e., *S. viridis*, *C. arvensis* L., *S. pumila*, *Avena fatua* L., *Papaver rhoeas* L., *A. vulgaris*, *Hyoscyamus niger* L., *Oxalis acetosella* L., *Abutilon theophrasti* Medik., and *Geranium pusillum* L. The following weed species disappeared completely from the records: *P. amphibium* L. Gray, *S. arvensis*, *Malva neglecta* Wallr., *Sisymbrium officinale* L. Scop., *Euphorbia helioscopia* L., and *Rumex acetosella* L.

Maize is also very vulnerable to weed infestation due to its slow growth in the first 4–6 weeks, low density per square metre (4–7 plants), and wide spacing between rows [25]. In 2020, global maize production amounted to about 1162.4 million tonnes (Americas—50.1%, Asia—31.4%, Europe—10.7%, Africa—7.7%, and Oceania—0.1%). The largest maize producers in Europe were: Ukraine (30.3 million tonnes), Russia (13.9 million tonnes), and France (13.4 million tonnes) [12]. Between 1973 and 1976, and again between 2002 and 2010, Fried et al. [26] compared the abundance and frequency of occurrence of segetal plants on maize plantations in 175 and 484 fields, respectively, located throughout France. They showed that the dominant species, whose regional frequency in both the first and second stages of observation was 21.1–64.3%, were: *C. album*, *E. crus-galli*, *A. retroflexus*, *P. maculosa* Gray, and *P. lapathifolia* L. Delarbre. In 2002, Fried et al. [27] analysed the influence of 14 agroecological factors on the composition and diversity of weed communities on 694 maize plantations throughout France. They identified the following weed species: *A. arvensis*, *V. hederifolia* L., *Papaver rhoeas* L., *M. arvensis* L., *Juncus bufonius* L., *A. myosuroides*, *G. aparine*, *A. retroflexus*, *E. crus-galli*, *Calystegia sepium* L. R. Br., *Cynodon dactylon* L. Pers., *S. pumila* Poir. Roem. and Schult., *Digitaria sanguinalis* L. Scop., *Datura stramonium* L., *Reseda phyteuma* L., *Anagallis foemina* Mill., *Bromus sterilis* L., *A. majus*, *R. acetosella*, *Phytolacca americana* L., *Portulaca oleracea* L., *Ranunculus sardous* Crantz, *A. arvensis*, *R. phyteuma*, *G. tetrahit*, *Legousia speculum-veneris* L. Durande ex Vill., and *M. perforata* Merat.

According to FAO statistics [12], Poland is also an important maize producer (6.7 million tonnes per year). Gołębiowska et al. [28] analysed changes in weed infestation of maize plantations in various soil sites in south-western Poland between 1963 and 2013. The researchers observed that in the first period of their study (1963–1972), the dominant segetal plant specieswere: *C. album* L., *E. crus-galli*, *G. parviflora*, *E. repens*, *S. arvensis*, *C. arvense*, *A. arvensis* L., *P. lapathifolium* L., *F. officinalis* and *P. persicaria*, of which the cover index ranged from 213.5–2039.7. During the second research period (1973–1982), *A. retroflexus* increased in density (cover index from 137.2 in the first study period to 333.1 in the second period), whereas the number of perennial weeds decreased. Between 1983 and 1992 (third period of study), there was greater diversity in annual dicotyledonous species and a significant increase in the number of plants with phytosociological stability in growth stages II–V. In the fourth period of the study (1993–2002), the researchers observed an increase in the number of *A. vulgaris* L., *G. tetrahit*, and *Erigeron canadensis* L. In the final period (2003–2013), the researchers observed an increase in the density of *S. pumila* Poir. Roem. and Schult, *S. viridis* L. P. Beauv., *Solanum nigrum* L., and *Aethusa cynapium* L., as well as the emergence of new expansive species, such as *A. theophrasti*. Regardless of the

observation period, the dominant weeds were *E. crus-galli* and *C. album*, whose cover index ranged from 1444.9–2823.

### 2.2. Consequences of Weed Infestation of Farmlands

According to the FAO, the world population will have exceeded nine billion by 2050, when it is estimated that the demand for food will have increased by up to 98%. Therefore, it is necessary to increase plant production in order to satisfy this demand [29–31]. According to Kumar et al. [30], 40% of the damage to crops is caused by various species of weeds, pathogens, insects, and vertebrates. Elkhouly et al. [32] divided the losses in commercial crops into qualitative and quantitative. Losses from the former result from the reduced market value of crops and a lower content of important nutrients, while losses from the latter are caused by the lower yield per area unit, which results from low productivity. According to Majrashi [31], lower yields of commercial crops may be caused by abiotic and biotic factors. Abiotic factors include temperature, soil salinity, access to water, light, and nutrients. Biotic factors can be divided into three basic groups, i.e., pathogens (fungi, viruses, and bacteria), animal pests (insects, mites, nematodes, snails, rodents, birds, and mammals), and weeds (monocotyledons and dicotyledons).

In contrast to other pests, weeds can develop over a wide range of environmental conditions and are responsible for the greatest loss of yield [29–35]. According to Gharde et al. [34], differences in actual yield losses are also influenced by the type of crop, location, soil type, plant growth conditions, and weed development. Zohaib et al. [33] estimate that segetal plants decrease wheat yield by 25–30%. According to Singh et al. [36], the uncontrolled growth of weeds at the critical stage of wheat growth may reduce yield by up to 60%.

The uncontrolled growth of weeds may reduce the yield of sugar beet crops by as much as 90–95% [37,38]. Bruciene et al. [39] and Abd El Lateef et al. [40] indicate that segetal plants reduce sugar beet production by 26–100%. According to Jain et al. [41] and Pant et al. [42], weed infestation decreases the yield of maize by 60–85%. Simon et al. [43] estimated that the loss of maize yield caused by weeds was as great as 90%. Kołodziejczyk et al. [44] and Zarzecka et al. [45] indicated that segetal plants reduced the production of potato (*Solanum tuberosum* L.) by 20–80%.

According to Li et al. [46], weeds not only reduce the quantity and quality of the yield but also increase labour inputs and the costs of production (costs of cleaning and drying products). Moreover, some weed species are food sources for pests that include pathogens and parasites of crops, and thereby contribute to their development [31]. Segetal plants also produce poisonous substances [47] and cause allergies in humans and animals [48].

According to Pszczółkowski et al. [49], the segetal flora is a permanent element of the agricultural biocenosis and, therefore, cannot be completely eliminated. The authors suggest that it should be limited to a level that will not generate large losses in commercial crops. However, segetal plantscan also have a positive influence on the ecosystem. They are involved in the cycling and balance of nutrients [19] and prevent soil erosion. Weeds can be habitats for beneficial insects and microorganisms, and, thus, increase the biodiversity of the microfauna and microflora [47].

### 2.3. Mode of Action of Weeds

Weeds compete with crops for environmental resources, such as space, nutrients, sunlight, and water [18,19,29,40]. The rapid growth of weeds and the morphological and physiological modifications in their root systems ensure greater absorption of nutrients from the substrate. As a result, they can be very competitive with crops [19,50]. According to Kocira and Staniak [47], the dominance of one or several weed species is more unfavourable to crops than a weed community with a highly diversified species composition. The growth and development of weeds depend on the crop cultivar [49,51,52], environmental conditions [27,49,53–55], and agricultural practices [13,19,44,49,51,56–59].

According to Sawicka et al. [19], segetal plants have numerous characteristics that enable them to develop in a hostile environment. They have adapted to agrotechnical cycles; they produce very large numbers of seeds with increased viability that are easily dispersed by the wind. As a result, they can survive exposure to severe stress [50]. Karkanis et al. [60] identified the main adaptive mechanisms that enable weeds to survive droughts and high temperatures: reduced total leaf area, change in leaf orientation, the presence of an additional layer of hairs on the surface of the leaves, shortened life cycle, increased root to shoot ratio, development of a deep root system and early stomatal closure. According to Singh et al. [36], Bufford and Hulme [61] and Sharma et al. [62], weeds are characterised by a greater genetic diversity and phenotypic plasticity than other plant species. They are even capable of epigenetic adaptation.

Invasive weed species produce allelochemicals that disorder the basic physiological processes of crops and, thus, inhibit their growth and development. This key mechanism enables weeds to compete with crops for environmental resources [63]. Segetal plants produce very large amounts of allelopathic compounds (Table 1), which are released into the environment in a number of ways. These compounds can be dissolved in water and leached from specific parts of the plants. Allelopathic substances can also be released in decomposing plant residues that accumulate in the soil, which can then reach the target crops by exposure. Volatile chemicals are released into the atmosphere and are then absorbed by crops with rain, dew or vapor. Some compounds flow into the rhizosphere through the roots and are then taken up by plants that grow in the immediate vicinity [33,64–69].

**Table 1.** Allelopathic weeds and the allelochemicals that they produce.

| Weeds | Allelochemicals | References |
|---|---|---|
| *Ageratum conyzoides* L. | ageratochromene, benzoic acid, coumaric acid, essential oils, gallic acid, protocatechuic acid, sinapic acid | [65,70] |
| *Alternanthera philoxeroides* Mart. | coumaric acid, hydroxy-methoxybenzoic acid | [33,66,70] |
| *Alternanthera sessilis* L. | chlorogenic acid, ferulic acid, gallic acid, vanillic acid | [33,66,70] |
| *Ambrosia artemisiifolia* L. | caryophyllene, germacrene, limonene, pinene | [69,71] |
| *Ambrosia trifida* L. | carotane sesquiterpenes, essential oils, thiarubrines, thiophenes | [65] |
| *Artemisia annua* L. | artemisinin | [67] |
| *Avena fatua* L. | caffeic acid, chlorogenic acid, coumaric acid, ellagic acid, ferulic acid, hydroxy-benzoic acid, scopoletin, vanillic acid | [33] |
| *Bidens pilosa* L. | caffeic acid, coumaric acid, dimethoxyphenol, ethyl-benzenediol, eugenol, ferulic acid, hydroxybenzoic acid, protocatechuic acid, pyrocatechin, salicylic acid, vanillic acid, vanillin | [65] |
| *Bothriochloa laguroides* DC Herter | dodecane, farnesol, hexadecane, tetradecene | [33] |
| *Callistemon citrinus* Curtis | leptospermone | [67] |
| *Centaurea maculosa* Lam. | catechin | [67] |
| *Chenopodia strummurale* L. S. Fuentes, Uotila and Borsch | benzoic acid, coumaric acid, ferulic acid, vanillic acid | [33] |
| *Chenopodium album* L. | chlorogenic acid | [33] |
| *Chenopodium ambrosioides* L. | ascaridole, limonene, monoterpenes, sesquiterpenes, triterpenes | [33] |
| *Cirsium arvense* L. Scop. | caffeic acid, chlorogenic acid, coumaric acid, ferulic acid, hydroxybenzoic acid, vanillic acid | [33] |
| *Convolvulus arvensis* L. | caffeic acid, chlorogenic acid, cinnamic acid, coumaric acid, ferulic acid, hydroxybenzoic acid, protocatechuic acid, pyrogallic acid, resorcinol, salicylic acid, syringic acid | [33] |
| *Conyza canadensis* L. | catechol, gallic acid, syringic acid, vanillic acid | [67] |

**Table 1.** *Cont.*

| Weeds | Allelochemicals | References |
|---|---|---|
| *Conyza stricta* Wild. | chlorogenic acid, coumaric acid, ferulic acid | [33,70] |
| *Cyperus esculentus* L. | coumaric acid, ferulic acid, hydroxybenzoic acid, syringic acid, vanillic acid | [33] |
| *Cyperus rotundus* L. | alkaloids, catechol tannins, flavonoids, furochromones, glycosides, sesquiterpenes | [33,70] |
| *Echinochloa colona* L. | apigenin, chlorogenic acid, cinnamic acid, ferulic acid, protocatechuic acid, syringic acid | [70] |
| *Echinochloa crus-galli* L. P. Beauv. | acenaphthene, coumaric acid, benzoic acid, cinnamic acid, decanoic acid, phthalic acid, diethyl phthalate, dihydrokavain, ferulic acid, hydroxymandelic acid, lactones, fatty acids, myristic acid, phenols, stearic acid, steroids, vanillic acid | [33,65,67,70] |
| *Eclipta alba* L. | benzoic acid, coumaric acid, ferulic acid, vanillic acid | [33] |
| *Eichhornia crassipes* Mart. | dimethylcyclopentane, isocyanatoethyl acetate, propane amide | [33] |
| *Imperata cylindrical* L. P. Beauv. | chlorogenic acid, coumaric acid, isochlorogenic acids, scopoletin, scopolin, syringic acid, vanillic acid | [65] |
| *Ipomoea chirica* L. Sweet | cinnamic acid, methylphenyl benzoate | [72] |
| *Lantana camara* L. | coumaric acid, furano-naphthoquinones, flavonoids, iridoid glycosides, lantadenes, methylcoumarins, monoterpenes, phenylethanoid glycosides, salicylic acid, sesquiterpenes, triterpenes | [65,73] |
| *Lathyrus aphaca* L. | caffeic acid, coumaric acid, gallic acid, syringic acid | [33,68,74] |
| *Leonurus sibiricus* L. | caffeic acid, phenols | [67] |
| *Medicago polymorpha* L. | coumaric acid, hydroxy-methoxybenzoic acid, vanillic acid | [33,68,74] |
| *Melilotus indica* L. | caffeic acid, chlorogenic acid, coumaric acid, ferulic acid, gallic acid, hydroxy-methoxybenzoic acid, syringic acid, vanillic acid | [33,68,74] |
| *Mikania micrantha* Kunth | benzoic acid, lactic acid | [72] |
| *Nigella sativa* L. | carvacrol, dithymoquinone, hederin, nigellicine, nigellidine, thymohydroquinone, thymol, thymoquinone | [33] |
| *Oryza sativa* L. | momilactone | [67] |
| *Parthenium hysterophorus* L. | anisic acid, caffeic acid, cardiac glycosides, coronopilic acid, coronopilin, coumaric acid, chlorogenic acid, ferulic acid, hydroxybenzoic acid, lactones, myrcene, ocimene, parthenin, pinene, saponins, steroids, tannins, vanillic acid, volatile compounds | [33,65] |
| *Piper longum* L. | sarmentine | [67] |
| *Polygonum barbatum* L. | acetophenone, caffeic acid, chlorogenic acid, coumaric acid, sitosterol | [33,70] |
| *Ruta graveolens* L. | cadinene, camphene, caryophyllene, cineol, copaene, cymene, decanol, decanone, decyl acetate, dodecanone, dodecene, eudesmol, flavonoids, furocoumarins, heptadecane, heptanone, hexadecane, hexadecanol, humulene, limonene, linalool, methyl salicylate, nonanol, nonanone, nonene, octanoicacid, octanol, octyl acetate, pentadecanol, pentadecanone, phenylethyl alcohol, pinene, terpinolene, tridecane, tridecanone, undecanol, undecanone, valeric acid, xanthotoxin | [33] |
| *Sambucus nigra* L. | cyaanogenins, flavonoids, lignans, phenolic glycosides | [67] |
| *Sorghum bicolour* L. Moench | sorgoleone | [67] |

**Table 1.** *Cont.*

| Weeds | Allelochemicals | References |
|---|---|---|
| *Sorghum halepense* L. Pers. | chlorogenic acid, coumaric acid, hydroxybenzaldehyde, phenolic compounds | [33] |
| *Sphenoclea zeylanica* Gaertn. | epi-zeylanoxide, zeylanoxide | [33] |
| *Stauranthus perforates* Liebm. | asarinin, fargesin, furanocoumarins, pellitorine, pyranocoumarins, sesquiterpene | [33] |
| *Stellera chamaejasme* L. | chamaejasmenin, daphnodorin, dihydrodaphnodorin, genkwanol, mesoneochamaejasmin, | [33] |
| *Terminalia catappa* L. | coumaric acid, ferulic acid, palmitic acid, pentadecanone, stearic acid, syringic acid, vanillic acid, β-sitosterol-glucoside | [33] |
| *Trigonella polycerata* M. Bieb. | coumaric acid, hydroxy-methoxybenzoic acid, syringic acid | [33,68,74] |
| *Vicia sativa* L. | coumaric acid, hydroxy-methoxybenzoic acid, ferulic acid | [33,68,74] |

Allelopathic weeds have a negative influence on crops and cause large losses on plantations. The compounds released by segetal plants limit the growth and development of the accompanying plants but also successive plants. They eliminate proteins of the pentose phosphate pathway and respiratory enzymes, thus reducing the vigour of seeds and inhibiting their germination. Moreover, allelopathic compounds produced by weeds delay the emergence of crop seedlings and they decrease the number and weight of sperm and seeds. They also inhibit the synthesis of sugars and proteins, which are important elements of the metabolic pathways in plant tissues. Allelochemicals also reduce the activity of antioxidative enzymes, but they increase the production of reactive oxygen species in plants. They also destroy chlorophyll cells, thus interrupting photosynthesis and cell division. As a result, crops grow and develop more slowly, have shorter roots and shoots, smaller leaves and lower dry weight. All these changes result in a lower quality and quantity in the crop yield. In addition, the allelopathic compounds produced by the weeds affect the chemical properties of the substrate, especially the pH, electrical conductivity, concentrations of potassium and chlorine ions, and the availability of soil nutrients [64,66,68,70].

## 3. Chemical Herbicides

### 3.1. Chemical Herbicides Available on the European Market

Herbicides are chemicals composed of organic compounds that are designed to limit the development of segetal plants or eliminate them completely. Herbicides contain active compounds that have the greatest effect on weeds, and a filler, which increases the effectiveness of herbicides and facilitates their use [18,35,75].

Chemical herbicides may contain one or more active substances. According to the Register of Authorised Crop Protection Products of 2022 [76], the following substances can most often be found in herbicides: 2,4-D, MCPA, aclonifen, amidosulfuron, aminopyralid, beflubutamid, benfluralin, bensulfuron-methyl, bentazon, bifenox, bromoxynil, quinoclamine, quinmerac, quizalofop-P-ethyl, quizalofop-P-tefuryl, clomazone, clopyralid, chlortoluron, chloridazon, cycloxydim, diflufenican, dicamba, dimethachlor, dimethenamid-P, ethofumesate, phenmedipham, fenoxaprop-P-ethyl, flazasulfuron, florasulam, fluazifop-P-butyl, flufenacet, flurochloridone, fluroxypyr, fluroxypyr ester, fluroxypyr-meptyl, foramsulfuron, glyphosate, glyphosate isopropylamine salt, halauxifen methyl, haloxyfop-p-methyl, indoxacarb, isoxaflutole, iodosulfuron-methyl-sodium, carfentrazone-ethyl, clethodim, clodinafop, lenacil, mecoprop-P, metamitron, metazachlor, metobromuron, metolachlor-S, metribuzin, metsulfuron-methyl, mesosulfuron-methyl, mesosulfuron, mesosulfuron-methyl, mesotrione, napropamide, nicosulfuron, oxyfluorfen, pendimethalin, pethoxamid, picloram, pinoxaden, pyroxsulam, pyridate, propaquizafop, propoxycarbazone-sodium, propyzamide, prosulfocarb, rimsulfuron, sulcotrione, tembotrione, terbuthylazine,

thiencarbazone-methyl, thifensulfuron-methyl, tribenuron-methyl, triflusulfuron-methyl, tritosulfuron, nonanoic acid, acetic acid, and pelargonic acid. Due to the mode of action and toxicity of active substances, some registered herbicides are classified as hazardous to the environment and bees, and harmful to human health.

The European Statistical Office distinguishes seven classes of herbicides, i.e., herbicides based on phenoxy-phytohormones, herbicides based on triazines and triazinones, herbicides based on amides and anilides, herbicides based on carbamates and bis-carbamates, herbicides based on dinitroaniline derivatives, herbicides based on derivatives of urea, uracil or sulfonylurea, and herbicides based on other compounds. According to Eurostat statistics [77], the largest group of herbicidal pesticides sold in 2020 were herbicides based on amides and anilides.

According to the Register of Authorised Crop Protection Products prepared by the Ministry of Agriculture and Rural Development on 13 April 2022 [76], 1082 herbicides are registered for use in Poland. The FAO statistics [12] show that the use of herbicides has increased both in Poland and the other EU memberstates in recent years. The average consumption of herbicides in EU memberstates amounted to 115,897 tonnes (in Poland—11,256 tonnes) in 2018, 116.780 tonnes (in Poland—11.371 tonnes) in 2019 and 120.112 tonnes (in Poland—11,685 tonnes) in 2020.

Nearly 45% of all crop protection products sold globally are chemical herbicides. The main producers of chemical herbicides are Canada, United States and China [30]. According to the European Statistical Office [77], sales of herbicides in the EU in 2020 amounted to 134,201 tonnes. The largest producer of herbicides was France (29,155 tonnes), whereas Poland was ranked fifth in terms of sales (12,809 tonnes).

*3.2. Mode of Action of Chemical Herbicides*

Herbicides can be divided into two basic groups according to the degree of selectivity. Selective herbicides affect only a specific weed species and do not affect other plants. In contrast, non-selective preparations attack several plant species. Moreover, three main groups of herbicides are applied directly into the soil. The first group includes substances applied into the substrate before the crops are planted. The second group includes pre-emergence preparations, which inhibit the germination and growth of weed seeds and are applied before the crops start to yield. The third group includes post-emergence agents, which are applied after the crops have started to yield [78,79].

There is no uniform classification of chemical herbicides according to their mode of action. Sherwani et al. [79] divided herbicides into 11 basic groups. A few years later, Dayan et al. [80] proposed the division of herbicides into three main groups with numerous subgroups including crop protection products that affect specific elements of the plants. According to the international Herbicide Resistance Action Committee (HRAC) [81], there are currently 25 groups of chemical herbicides (Scheme 1).

Photosynthesis is the most important biochemical process as it is the main source of oxygen for all organisms on Earth. Herbicides that affect photosynthesis disrupt and change the direction of electron transport. This inhibits energy production and carbon dioxide fixation, and causes the accumulation of nitrites and the loss of carotenoids, chlorophyll and ascorbate. As a result, leaves lose their pigmentation and turn white or wilt and rapidly desiccate. Examples of herbicides that inhibit photosystem II include atrazine, diuron, propanil, bromoxynil, monuron, isoproturon, linuron, simazine, chloridazon, bromacil, terbacil, lenacil, phenmedipham, and metribuzin [75,78–80,82–85].

Diflufenican, flurtamone, fluridone, and norflurazon [85] are active substances of herbicides that reduce the activity of phytoene desaturase (PDS), which catalyses the production of carotenoids that are responsible for photosynthesis and giving colour to plants. Carotenoids are yellow, orange, red, and pink pigments that stabilise the chloroplast membrane, suppress the reactive oxygen species in chloroplasts, and accumulate chlorophyll. In addition, they protect the photosynthetic apparatus from photodegradation, i.e., the decay caused by an excessive amount of energy generated as a result of very intense

photosynthesis. The PDS inhibitors stimulate plant organisms to produce white tissues, thus causing organ necrosis [75,78–80,83].

---

**CLASSIFICATION OF CHEMICAL HERBICIDES**

**HERBICIDES AFFECTING LIGHT ACTIVATION OF REACTIVE OXYGEN SPECIES**

**Groups 5 and 6.** Photosystem II inhibitors
**Group 10.** Glutamine synthetase inhibitors
**Group 12.** Phytoene desaturase inhibitors
**Group 13.** Deoxyxylulose-5-phosphate synthase inhibitors
**Group 14.** Protoporphyrinogen oxidase inhibitors
**Group 22.** Photosystem I inhibitors
**Group 27.** 4-hydroxyphenylpyruvate dioxygenase inhibitors
**Group 32.** Solanesyl diphosphate synthase inhibitors
**Group 33.** Homogentisate solanesyl transferase inhibitors
**Group 34.** Lycopene cyclase inhibitors

**HERBICIDES AFFECTING CELL METABOLISM**

**Group 1.** Acetyl coenzyme A carboxylase inhibitors
**Group 2.** Acetolactate synthase inhibitors
**Group 9.** 5-enolpyruvylshikimate-3-phosphate synthase inhibitors
**Group 15.** Very-long-chain fatty acid synthase inhibitors
**Group 18.** 7,8-dihydropteroate synthase inhibitors
**Group 28.** Dihydroorotate dehydrogenase inhibitors
**Group 29.** Cellulose synthesis inhibitors
**Group 30.** Fatty acid thioesterase inhibitors
**Group 31.** Serine-threonine protein phosphatase inhibitors

**HERBICIDES AFFECTING CELL DIVISION AND GROWTH**

**Group 3.** Microtubule inhibitors
**Group 4.** Auxin herbicides
**Group 19.** Auxin transport inhibitors
**Group 23.** Microtubule organisation inhibitors
**Group 24.** Uncouplers

**Scheme 1.** Classification of chemical herbicides according to their mode of action, proposed by the Herbicide Resistance Action Committee [81].

Clomazone is a proherbicide that interacts with weed plants and is converted into ketoclomazone. It has the same effect as phytoene desaturase inhibitors. Due to this bioactive form, this herbicide can inhibit deoxyxylulose-5-phosphate synthase (DOXP), which controls the synthesis of terpenoids (isoprenoids) responsible for the taste and smell of plants [75,80,83,84].

Nitrofen, acifluorfen, butafenacil, and flumiclorac, contained in herbicides, inhibit protoporphyrinogen IX oxidase (PPO), which controls the synthesis of porphyrins [80,85]. This causes the accumulation and oxidation of chlorophyll and heme synthesis intermediates and the production of highly reactive oxygen species. These oxygen molecules

induce lipid peroxidation and degrade weed cell membranes [75,78,82]. As a consequence, plant pigments leak, and leaves stick and wrinkle, thereby causing plant necrosis and death [79,83,84].

According to Dayan et al. [80,83], herbicides contain active substances, such as dichlobenil, isoxaben, and indaziflam, which inhibit cellulose synthase and, thus, disrupt the production of cellulose microfibres in the cell walls of the weed plants. Dichlobenil immobilises cellulose synthase and increases its accumulation. Isoxaben completely destroys cellulose synthase, which inhibits the growth of plants and causes swelling of the roots.

Alachlor, flufenacet and allidochlor are herbicides that may disturb the action of VLCFA synthase [85]. This enzyme is responsible for the production of the very long-chain fatty acids contained in waxes, cutins and suberins, which maintain moisture and limit the transport of various compounds into cells. The inhibition of this enzyme inhibits plant division and growth, and causes leaf curling or twisting [75,79,80,83,84].

Tryptophan, tyrosine and phenylalanine are aromatic amino acids produced by enolpyruvyl shikimate-3-phosphate synthases (EPSPS). The active substance in glyphosate acts as a transition state analogue of the phosphoenolpyruvate substrate, making it the only inhibitor of the EPSPS enzyme [75,79,82,84,85]. Glyphosate inhibits the synthesis of aromatic amino acids, and increases the accumulation of shikimate and the flow of carbon and phosphates to the shikimate pathway. This disturbs the carbon-binding process and causes changes in chlorophyll fluorescence [80,83].

Glufosinate inhibits glutamine synthetase (GS) [85], the enzyme that converts ammonia and glutamic acid into glutamine [83]. GS inhibition increases the accumulation of ammonia and reduces the pH gradient on both sides of the plant cell membrane [75,79]. As a result, the synthesis of amino acids is inhibited by a reduction in the amount of glutamine and glutamate and by limiting transporters, which is manifested by the wilting of leaves [80].

Dayan et al. [80,83] observed that active substances, such asquinclorac, aminocyclopyrachlor and picloram, act as natural hormones. They increase plant metabolism and accelerate growth, which results in increased synthesis of ethylene and abscisic acid (ABA). High ABA concentrations lead to the closure of the stomata, disrupt carbon dioxide assimilation, and result in the formation of reactive oxygen species. These active substances cause swelling of plant tissues and the curling of leaves and stems. Diflufenzopyr and napalam are examples of herbicides that slow down plant growth by inhibiting the transport of auxins responsible for the apical dominance of the main shoot, floral meristem differentiation, and correctplant growth and development. Asulam is the only preparation that disturbs the synthesis of folic acid. It causes chlorosis, inhibits the growth of new tissues, and causes ageing in older tissues [80,83].

The effectiveness of chemical herbicides depends on the way they are applied, absorbed, transported and degraded in the weed plants [51]. The essential determinants of the effectiveness of these agents are the weed species and phase of development, as well as environmental conditions, such as temperature, humidity, and carbon dioxide level [86]. According to Varanasi et al. [54], any change in climatic conditions will affect plant physiology, which might adversely influence the effectiveness of herbicidal preparations.

### 3.3. Mechanisms of Weed Resistance to Herbicides

As a consequence of repeated use of the same chemical preparations, segetal plants have developed strong resistance mechanisms to all possible herbicidal modes of action, which they pass on to the next generation [82,87]. According to Heap et al. [88], many plants have developed cross resistance (one mechanism results in resistance to several herbicides) [89–91] or multiple resistance (several resistance mechanisms) [92–94]. Currently, the greatest problem is the control of weed species that are resistant to several herbicides with different modes of action.

Since 1975, the insensitivity of segetal species to chemical herbicides has increased sharply, and it is estimated that there are 512 cases of weed resistance to commercially available herbicides worldwide, including 380 segetal taxa [95]. The greatest number of

weeds resistant to chemical herbicides have been recorded on wheat plantations (74 species of segetal plants). The most insensitive weeds that have developed multiple resistance are *L. rigidum* Gaud., *P. annua*, *A. palmeri* S. Wats., *E. crus-galli*, *Eleusine indica* L. Gaertn., *L. perenne* L., *A. tuberculatus* Moq. J. D. Sauer, *A. fatua*, *A. hybridus* L., and *Raphanus raphanistrum* L.

Plants exhibit two basic mechanisms of resistance to herbicides, i.e., TSR (target-site resistance) and NTSR (non-target-site resistance) [82,96]. Usually, one type of resistance predominates, although researchers have identified an increasing number of cases of interactions between these two mechanisms [97–99]. According to Gaines et al. [100], plants develop combined insensitivity mechanisms as a result of cross-pollination.

Target-site resistance is induced in two ways. The first is a DNA mutation, which is the substitution of one or more amino acids in the target enzyme. This causes changes in the protein structure, which prevents the binding of the herbicide. The other method of inducing resistance is the overexpression of the target enzyme, which is caused by the deletion of codons, a change in the gene promoter or the amplification of the gene coding the biosynthesis of the protein to which a specific herbicide then binds [100–102]. Zakaria et al. [103], Yanniccari et al. [104] and Zhang et al. [105] observed that the point substitution in acetohydroxy acid synthase, enolpyruvate-shikimate-3-phosphate synthase, and glutamine synthetase, which are responsible for the biosynthesis of valine, tryptophan, tyrosine, or phenylalainine, has made species, such as *Limnocharis flava* L. Buchenau, *D. sanguinalis* and *E. indica*, insensitive to glyphosate and glufosinate. According to Figueiredo et al. [106], *S. orientale* developed resistance to 2,4-dicholorophenoxyacetic due to a deletion in the indole-3-acetic acid coreceptor. Silva et al. [107] and Widianto et al. [108] observed that mutations in the acetolactate synthase gene in *E. sumatrensis* Retz. and *Monochoria vaginalis* Burm. f. C. Presl. resulted in the development of resistance to inhibitors of the ALS 9 enzyme (acetolactate synthase). Gherekhloo et al. [109] found that different resistance mechanisms at a target site could combine with each other to produce increased insensitivity to specific herbicides. Their research showed that the combination of mutations in the 5-enolpyruvylshikimate-3-phosphate synthase gene and the overexpression of this enzyme in *E. indica* resulted in its strong resistance to glyphosate.

Delye et al. [110] identified the following factors that influence the development of resistance at the target site: the rate of mutation, the flow of genes between populations, the degree of inheritance, and the initial frequency of occurrence of resistance alleles in the weed species.

NTSR is controlled by various genes. This enables weeds to become insensitive to several preparations with different modes of action. The primary task of the processes that occur within NTSR is to minimise the concentration of the herbicide at the target site [111,112]. There are three major mechanisms of this resistance: increased metabolism, decreased herbicide absorption, and altered translocation patterns [95,113]. There are two phases involved in the metabolism of herbicides: the activation phase and the conjugation phase. The activation phase involves the hydrolysis, reduction or oxidation of non-polar particles of herbicides [100]. Hydrophilic, water-soluble compounds are formed, which are ready for further detoxification. This phase inhibits the accumulation of toxic substances and the distribution of herbicides in the plant [113]. The results of the research by Oliveira et al. [114], Yang et al. [115] and Guo et al. [116] clearly showed that cytochrome P450 is the enzyme responsible for increased metabolism. In the second phase of metabolism, herbicide molecules are inactivated and detoxified, which makes them less harmful. The main enzymes during this step are GSH S-transferases (GST) or glucosyltransferases (GT), which conjugate the active compounds to GSH or glucose, respectively. These transformations result in the formation of conjugates, which are soluble in water and separated by membrane transport proteins [111,113].

The alteration of the herbicide translocation patterns in the plant is another mechanism to develop NTSR to chemical herbicides. Reduced translocation may be induced by sequestration, root exudates, hypersensitivity reactions, or impaired transport. As a result of these mechanisms, active substances are retained in the weed plants in a part of the

plant that is not responsible for growth and development. The basic mechanism of weed resistance to paraquat, for example, is the retention of the herbicide in the protoplasm and its sequestration in the vacuoles. Glyphosate also undergoes sequestration in the vacuoles. This process is controlled by active transport with a membrane transporter [117]. This herbicide inhibits the activity of the EPSPS enzyme (5-enolpyruvyl shikimate-3-phosphate synthase), which causes the necrosis and death of mature leaves, possibly due to the increased production of hydrogen peroxide [111,113]. Another cause of reduced translocation of herbicidal preparations is the exudation of various substances from the roots of the weed plants, which are stimulated by biotic and abiotic stress. Resistant species transport large amounts of herbicides to the underground parts of plants very quickly. The exudation process can be passive or active. Active transport through cell membranes requires energy in the form of adenosine triphosphate (ATP) and the use of special transporters, while the passive secretion of herbicides takes place through vesicular transport, diffusion or ion channels [113].

Variable temperature, carbon dioxide levels, and humidity are the main environmental factors that affect the NTSR mechanisms. The degree of their influence on the weed plants depends on the plant physiology and the mode of action of the herbicides [82].

Active substances from herbicides enter plants and initiate a series of reactions within. They cause epigenetic changes in the plant genes that regulate resistance to environmental stressors, such as herbicides. Epigenetic processes involve chemical changes in the plant DNA and histones that may be induced by environmental stress, occur spontaneously, or be genetically passed down from generation to generation through mitosis and meiosis [118–121].

### 3.4. Effect of Chemical Herbicides on the Environment and Human Health

Over 90% of chemical crop protection products do not reach the target organisms [122]. The mode of action of these toxins depends on the species of the organism and the type of tissues exposed [123].

Herbicides can be applied to the substrate or directly to segetal plants. The transporters and channels contained within the weed plants distribute the active substances in the herbicidal preparations to all organs, including the roots, from where they can be excreted outside the plant. Active compounds pass through the rhizosphere and accumulate in the soil. Consequently, they can pass to the roots of other plants and come into contact with soil organisms. Some herbicides evaporate from the surface of weeds and from the substrate. When they enter the atmosphere, air currents can transfer them to other areas [35]. Rain and surface runoff wash harmful compounds into water systems, where they then undergo sedimentation or suspension [122,123]. El-Nahhal [122] and Wan et al. [124] have reported that several types of herbicides had also been detected in drinking water.

The skin, respiratory system and oral cavity are the direct routes for pesticides to enter human and animal bodies. Chemically active compounds are also taken up indirectly through the food chain, whereby plants or animals that contain active substances are eaten by other organisms [35,125–127].

Excessive concentrations of herbicides in the environment may disturb the biological balance in the soil, reduce its abundance and fertility, and consequently lead to its degradation [128]. Research has shown that active substances in herbicides disturb the synthesis of nitrites, nitrogen fixation, and the mycorrhiza process. As a result of these disturbances, microbial populations produce persistent compounds that are more harmful and toxic than the pesticides themselves [35].

Invertebrates are the most sensitive to water contamination with pesticides, whereas fish have a greater capacity to accumulate chemicals [123]. Herbicides increase the reproductive toxicity of aquatic species. Their consumption is a life and health hazard for other organisms, including humans. Research has shown that active substances in herbicides limit the growth and development of aquatic plants; the oxygen content in the water declines and the animal population declines [35]. Marin-Morales et al. [128] observed that

excessive concentrations of toxic compounds resulted in a loss of symbiotic interactions between algae and corals, which led to the bleaching of coral reefs. In aquatic organisms, herbicides have been shown to inhibit enzymatic transformations, and the impairment of nerves, eyes, brain, kidneys and the liver. Moreover, herbicides decrease immunity, cause changes in the blood composition, reduce food intake, create swimming disorders and lead to deformed skeletons in fish. In addition, they disorder the function of the reproductive system, embryonic development, reproduction, hormonal balance, circulatory system, metabolism and the DNA structure in cells [35,123,125,127,129–131].

Cullen et al. [132], Zioga et al. [133] and Christen et al. [134] observed that chemical crop protection products have been a key factor in the reduction in pollinating insect populations, especially bees. These animals are exposed to harmful substances mainly through direct contact with the preparation, contaminated pollen or nectar. In the case of the human organism, active substances in herbicides disrupt the DNA structure; hormonal balance; metabolism; the function of blood components; nervous, muscular and reproductive systems; and are responsible for a decrease in human and animal fertility levels [122,125,134–137]. According to Van Bruggen et al. [125] and El-Nahhal et al. [122], chemical crop protection products are carcinogenic. Many researchers have observed that increased exposure to herbicides increases the risk of non-Hodgkin lymphoma [138], multiple myeloma, breast cancer, colorectal cancer and skin cancer [139]. Exposure to herbicides also increases the risk of kidney disease, Alzheimer's disease, Parkinson's disease, ADHD and autism [125,140,141].

## 4. Bioherbicides

### 4.1. Mode of Action of Microorganisms on Weeds

Due to the withdrawal of various chemical herbicides from the market, manufacturers of crop protection products have sought to identify bioherbicides (environmentally friendly biological agents that are not toxic to humans and animals) in order to ensure the health and stable yield of crops.

Bioherbicides are natural herbicides made from plant extracts but also from living organisms (bacteria, fungi, and viruses) or their secondary metabolites produced during their growth and development [30,142]. Currently, the use of viruses as bioherbicides is minimal due to their high genetic variability and unstable host specificity [143].

Bioherbicides are composed of the bacteria and fungi that infect weed plants and thus inhibit seed germination and plant growth [30,142–144]. The defence mechanism of a plant is a key factor in the pathogen–weed relationship. Microbes have various virulence factors that enable them to overcome the resistance barriers produced by the weed plants and fully infect the target plant. There are two main groups of virulence factors. The first group includes enzymes that degrade cell walls, lipid membranes and weed proteins. This group includes amylases, pectinases, cellulases, lignin-modifying enzymes, proteases, peptidases and phospholipases. The second group of virulence factors include peptides and secondary metabolites with phytotoxic properties that disorder the physiological and metabolic processes in weeds. Phytotoxins, such as hydrogen cyanide (HCN), ethylene, ammonia, dimethyl disulphide, indole-3-acetic acid, hydrocinnamic acid and aminolevulinic acid [144] inhibit the function of intracellular enzymes and directly or indirectly influence gene expression, thus causing the death of the host organism [142].

Allelopathic bacteria produce phytotoxins and antibiotics that degrade weed plant cell walls and membranes. These metabolites are characterised by high specificity to segetal species and low toxicity to non-target organisms. In addition, rhizobacteria induce the expression of genes responsible for immunity, which increase crop tolerance to biotic and abiotic stresses [145]. According to Phukan et al. [146], the task of allelopathic bacteria is not to cause disease and eliminate weed plants completely, but to inhibit their early growth and development and reduce their competitiveness against crops.

Microorganisms produce phytotoxins with a herbicidal effect, such as HCN. This toxic gas can formmetal complexes with the functional groups of various intracellular

enzymes. A high HCN concentration disrupts the uptake of nutrients by the plant, the assimilation of carbon dioxide and nitrates, and the transport of electrons in photosynthesis and the respiratory chain [144,146]. *Pseudomonas* is one group of bacteria that can produce significant amounts of HCN and, thus, limit weed growth [146]. Lakshmi et al. [147] observed that the KC1 strain of *P. aeruginosa* bacteria reduced the length of shoots of *A. spinosus* L. and *P. oleracea* (by 31.17–75.85% under laboratory conditions) and the length of roots (by 19.26–89.3% under laboratory conditions). This effect was caused by the significant amounts of HCN (from 4.78 to 6.98 nmol/L in the presence of glycine) produced by this bacterial strain. The researchers also found that gaseous metabolites limited weed growth more effectively than liquid compounds and that HCN had a greater effect on weed seedlings than on wheat seedlings. Lawrance et al. [148] isolated the H6 strain, which inhibited the germination and growth of *Cenchrus purpureus* Schumach. Morrone, *Oryza sativa* L., *Pisum sativum* L., and *A. spinosus*. The isolated strain not only produced HCN but also produced ammonia, siderophore, choline and its derivatives, which influenced its herbicidal effect. The inoculation of plants with the H6 strain distorted their roots and leaves, caused brown spots, inhibited the growth of root hair, and resulted in local necrosis. Flores-Vargas et al. [149] observed that the *P. fluorescens* was also capable of producing considerable amounts of HCN. This metabolite limited the growth of wild radish cultivated in a greenhouse and caused its chlorosis.

Rhizobacteria also secrete ethylene, an excessive concentration of which also limits the growth of weeds [146]. The herbicidal effect of volatile substances was observed by Zhao et al. [150], who found that the BMP-11 strain of *Paenibacillus polymyxa* bacteria produced octenol, benzothiazole, and citronellol, which disordered the germination and growth of *C. album*, *A. retroflexus*, and *E. crus-galli*. Verdugo-Navarrete et al. [151] observed that the volatile and diffusible metabolites produced by *Bacillus*, *Pseudomonas* and *Enterobacter* (in some cases by >50% 6 days after planting) were able to inhibit seed germination and reduce the length of *A. palmeri* S. Wats. seedlings (by 67.1–92.9% within 6 days of planting), roots (by 28.6–42.3% within 2 weeks of planting) and shoots (by 0–22.1% within 2 weeks of planting). Park et al. [152,153] found that the I-4-5 and I-3 strains of *Enterobacter* inhibited plant growth by regulating the synthesis of abscisic acid and disrupting the synthesis of gibberellins. The following year, the same group of researchers found that the I-3 strain of *Enterobacter* could be an effective alternative in the control of *Cyperus microiria* Steud. and *D. sanguinalis* [154]. Another group of compounds produced and secreted by microorganisms is phytotoxic metabolites, which disrupt the synthesis of macroparticles and damage the cell wall of weed plants [146]. For example, the WH6 strain of *P. fluorescens* produces such compounds. This isolate releases the primary germination-arrest factor (GAF), which arrests weed germination. The GAF disorders the function of the catalysts whose co-factor is pyridoxal phosphate, i.e., enzymes of nitrogen metabolism and ethylene biosynthesis. The WH6 strain secretes the primary herbicidal factor that inhibits the sprouting of a wide range of segetal plant species [143]. The D7 strain of *P. fluorescens* secretes a complex of peptides and fatty acid esters contained in the lipopolysaccharide matrix and, thus, inhibits the growth of *B. tectorum* L. roots [155]. Another group of compounds secreted by bacteria are exopolysaccharides (EPS), which colonise plant tissues and disrupt the transport of water through xylem vessels, and, thus, cause various plant species to wilt. For example, the JT-P482 strain of *Xanthomonas campestris* bacteria produce polysaccharide substances, which caused a large number of annual Kentucky bluegrass plants to wilt [146]. Another substance produced by rhizobacteria is aminolevulinic acid (ALA). This compound is an intermediate metabolite in the biosynthesis of tetrapyrroles, including chlorophyll, porphyrin and heme. An increased concentration of ALA causes the accumulation of chlorophyll intermediates, which are photosensitisers for singlet oxygen production. This results in the photodynamic damage and infection of the plant [156]. Phour et al. [157] found that eight strains of rhizobia were capable of producing large amounts of ALA (more than 10 ug/mL), which inhibited the germination of *Lathyrus aphaca* L. and the growth of plant organs. The JMM24 strain of *Bacillus flexus* limited the development of this plant most

effectively (92% decrease in root dry matter and 37% decrease in shoot dry matter within 75 days of planting).

Fungi also inhibit the growth and development of weeds by producing specific groups of secondary metabolites [158,159]. Fungal phytotoxins infect plants, disorder the correct functioning of their systems and, consequently, cause the death of their host [160]. Depending on the species, pathogens can infect plants mainly through colonisation of their aboveground and underground organs. Microorganisms enter the host tissue through natural openings, such as stomata, or through mechanical damage. These pathogens degrade the cell wall of their hosts enzymatically or form special structures that are responsible for the penetration of cuticle cells. After colonisation of the host tissue, the microorganisms function as a biotroph and obtain the necessary nutrients from their host or as necrotrophs, leading to the death of plant cells [161].

Studies have shown that mycotoxins belong to naturally occurring compounds, such as amino acids, aromas, coumarins, isocoumarins, terpenes, cytochalasins, trichothecenes, phenols, steroids, xanthones, quinones, furandiones, terpenoids, depsipeptides, alkaloids, polyketones, flavonoids, benzopyranones, and tetralones [159,162]. Xu et al. [163] distinguished five major groups of phytotoxic secondary metabolites derived from fungal cells: polyketides, phenols, phenolic acids, terpenoids, nitrogen-containing metabolites, and other phytotoxic metabolites. Moreover, the authors [163] classified these compounds as host-specific (selective) and non-specific (non-selective) toxins. According to this definition, selective metabolites cause symptoms only in the plants that host them and provide them with a suitable environment for their development. These compounds are mainly produced by pathogenic fungi, such as *Alternaria*, *Colletotrichum*, *Helminthosporium*, and are responsible for their pathogenicity. Non-selective metabolites affect not only their host, but also other plant species. They are not necessary to maintain the pathogenicity of their producers, but only affect their virulence and determine the range of the host.

Endophytes that colonise healthy plant tissues are a special example of microorganisms that produce fungal phytotoxins. Endophytic fungi remain dormant in the host organism for most of their lives until the emergence of conditions that favour the development of their pathogenic form and the production of toxic secondary metabolites [159].

According to Triolet et al. [164], the following fungi produce the most herbicidal phytotoxins: *Fusarium* sp., *Phoma* sp., *Penicillium* sp., *Alternaria* sp., *Ascochyta* sp., *Paraphoma* sp., *Rutstroemia* sp., *Drechslera* sp., *Diaporthe* sp., *Phyllosticta* sp., *Curvularia* sp., *Pyrenophora* sp., *Stemphylium* sp., *Myrothecium* sp., and *Gliocladium* sp. (Table 2).

**Table 2.** Herbicidal phytotoxins produced by fungi.

| Genus | Phytotoxins | References |
|---|---|---|
| *Alternaria* sp. | AAC-toxin, AAL-toxin, alternethanoxin, maculosin, tentoxin, tenuazonic acid, vivotoxin, vulculic acid | [160,164–167] |
| *Ascochyta* sp. | agropyrenal, agropyrenol, agropyrenone, ascaulitoxin, ascosonchine, cyperin, trans-aminoproline, triamino-hydroxyoctanoic acid | [160,168] |
| *Aspergillus* sp. | tenuazonic acid | [166] |
| *Colletotrichum* sp. | colletochlorin, dirhamnolipid, orcinol, tyrosol | [169] |
| *Curvularia* sp. | butyl isobutyl ester, dehydrocurvularin, phthalic acid, radicin | [160,164] |
| *Diaporthe* sp. | gulypyrone, hydroxybenzaldehyde, methylbenzoic acid, nectriapyrone, nitropropionic acid, phomentrioloxin, succinic acid | [160,164] |
| *Drechslera* sp. | drazepinone, ophiobolin, pyrenophorin | [160,164] |
| *Edenia* sp. | preussomerin, palmarumycin | [160] |
| *Fusarium* sp. | beauvericin, decalin, dehydrofusaric acid, diacetoxyscirpenol, enniatin, fumonisin, fusaric acid, moniliformin, radicin, rhodolamprometrin, tetracides, trichothecenes, zearalenone | [160,164,170] |

**Table 2.** *Cont.*

| Genus | Phytotoxins | References |
|---|---|---|
| *Gliocladium* sp. | viridiol | [164] |
| *Myrothecium* sp. | roridin | [164] |
| *Paraphoma* sp. | curvulin, phaeosphaeride | [164,171] |
| *Penicillium* sp. | cinnamic acid, dihydrosporogen, hydroxybenzoic acid, isopetasol, protocatechuic acid, salicylic acid, sporogen, vanillic acid | [160,164] |
| *Phoma* sp. | chenopodolans, cyperin, cytochalasins, desoxaphomin, herbarumin, hydroxybenzaldehyde, hydroxymelein, macrocidins, nitrophthalic acid, phomachalasin, putaminoxin, spirostaphylostrychnine, tenuazonic acid | [164–168,171–173] |
| *Phomopsis* sp. | nonenolide, phomentrioloxin | [160] |
| *Phyllosticta* sp. | phyllostictine, scytolide | [160,164] |
| *Preussia* sp. | cyperine | [160] |
| *Pyrenophora* sp. | papyracillic acid, pyrenophorin, triticone | [160,164] |
| *Pyricularia* sp. | tenuazonic acid | [160] |
| *Rutstroemia* sp. | ethylfusarubin, terpestacin | [164] |
| *Scytalidium* sp. | scytolide | [160] |
| *Stagonospora* sp. | modiolide, stagonolide | [160] |
| *Stemphylium* sp. | pyrenophorin | [164] |

While allelopathic bacteria produce phytotoxic compounds, fungal phytotoxins are responsible for the production of reactive oxygen species, lipid peroxidation, and changes in the composition of amino acidsin weed plants. They inhibit the activity of specific enzymes, disorder electron transport and cell division. They destroy cell organelles, inhibit the synthesis of chlorophyll and other important substances. Herbicidal fungi inhibit the growth and development of seedlings, reduce the biomass of plants, and cause intense chlorosis and the bleaching of the green parts of the plants, necrosis, and brown spots on leaves, which consequently cause the weed plants to wilt [160,165,166,174,175].

*4.2. Advantages of Bioherbicides*

Bioherbicides based on microorganisms are an alternative to chemical herbicides. They are very selective and specific, which means that they do not affect crops growing near segetal weeds. Bioherbicides are less toxic and less harmful to the environment and human health than chemical herbicides [176,177]. As the half-life of bioherbicides is shorter than the half-life of chemical substances [142], they decompose relatively rapidly, do not accumulate in the environment and do not pollute ecosystems [176]. According to Hasan et al. [178], some natural allelochemicals dissolve very easily in water and do not require chemical surfactants. The microorganisms used for the production of bioherbicides proliferate quickly and are widely available, which translates into lower production costs. Even small doses of bioherbicides can effectively inhibit the growth of weeds [48]. Moreover, bioherbicides eliminate effectively the segetal species that have developed resistance to chemical herbicides [176]. According to Bordin et al. [177] and Hasan et al. [178], natural phytotoxins have various modes of action, which reduce the risk of resistance. In addition, the use of bioherbicides is very useful for organic farming [179].

An abundance of research confirms that preparations based on biological agents are as effective as, and sometimes even more effective than, chemical herbicides. According to Zhang et al. [180], the SC64 strain of the fungus *Sclerotium rolfsii* largely eliminated the invasive *Solidago canadensis* L., thus increasing the biodiversity of the habitats in which this species was present. The authors proved that mycoherbicide reduced weed density

by 90.1% in the spring and by 79.1% in the autumn, while chemical glyphosate reduced the density of the studied species by 73.4% in the spring and by 66.4% in the autumn. In addition, the study showed that the microbial preparation was more stable and constant 80 days and 190 days after the treatment and it showed greater activity and ability to eliminate weeds (mortality rates of 92.1% and 89.6%, respectively) compared to the synthetic herbicide (mortality rates of 76.6% and 70.1%, respectively). The elimination of the invasive species *S. canadensis* allowed for the development of other plants, thus leading to an increase in the species diversity of habitats and the rapid restoration of the local plant community structure. Raza et al. [181] showed that the combination of allelopathic rhizobacteria and allelopathic water sorghum extract not only reduces the growth of invasive weeds (a reduction in seed germination by 38–50%) more effectively, compared to chemical agents, but also increases the wheat yield (seed germination by 86% in vitro) and the net benefits associated with its production. Their research estimated that the combination of these two biological weed control methods could bring about a significantly greater net benefit compared to control and chemical herbicides by 72.4% and 27.7%, respectively. In turn, the marginal rate of return (MRR) for the biopreparations used may amount to 4226.9%, compared to 52% for synthetic agents.

### 4.3. Commercially Available Bioherbicides

Bioherbicides are applied to inhibit the growth of weed plants and to eliminate them. They are a safe alternative to chemical herbicides. To date, these preparations have been used on plantations (fields and orchards) and in the natural environment (pastures and forests) [5].

The proportion of bioherbicides of all the pesticides produced worldwide is 1.3% (90% of biopesticides are insecticides). The International Association of Producers of Biocontrols (IBMA) estimates that the current market for biological pest control products is worth around EUR 550 million in Europe and EUR 1.6 billion worldwide [182]. The association also projects that the value of the biocontrol market will increase by 15% by 2050. According to Bordin et al. [177], the largest part of potential and still-analysed bioherbicides are preparations based on fungi (44%), plants (38%), bacteria (16%) and lichens (2%). In 81% of cases, the main factor of biocontrol is metabolites produced by microorganisms, and only 19% of cases are living cells of microorganisms.

To date, 25 bioherbicides have been registered globally. Most are fungicides (mycoherbicides) (Tables 3 and 4). Among the registered biopesticides, one is based on the tobacco mosaic virus. The following numbers of bioherbicides have been registered in the following countries: United States—11, Canada—9, Japan—2, South Africa—2, the Netherlands—1, Ukraine—1, China—1, India—1, and Africa–1 [30].

**Table 3.** List of registered bioherbicides based on bacteria.

| Name of Product and Manufacturer | Place and Year of Registration | Microorganisms | Target Weeds | References |
|---|---|---|---|---|
| **Albobacteryn** manufacturer unknown | Ukraine year unknown | *Achromobacter album* | various species | [5,30] |
| **Camperico** Japan Tobacco | Japan 1997 | strain JTP482 *Xanthomonas campestris* pv. *poae* | *Poa annua* L. | [5,30,142,183] |
| **Organo-Sol** manufacturer unknown | Canada 2010 | strain LTP-111 *Lactobacillus casei*, strain LTP-21 *Lactobacillus rhamnous*, strain LL64/CSL *Lactobacillus lactis* ssp. *Lactis*, strain LL102/CSL *Lactobacillus lactis* ssp. *Lactis*, strain M11/CSL *Lactobacillus lactis* ssp. *Cremoris* | *Trifolium repens* L., *Trifolium pretense* L., *Lotus corniculatus* L., *Medicago lupulina* L., *Oxalis acetosella* L. | [5,30,142,183] |
| **MBI-005 EP** Marrone Bio Innovations | USA Japan 2012 | thaxtomin A strain RL-110 *Streptomyces acidiscabies* | cotyledonous species | [5,30,142] |

**Table 4.** List of registered bioherbicides based on fungi.

| Name of Product and Manufacturer | Place and Year of Registration | Microorganisms | Target Weeds | References |
|---|---|---|---|---|
| **Acremonium diospyri** manufacturer unknown | Canada 1960 | *Acremonium diospyri* | *Diospyros virginiana* L. | [30,183] |
| **Lubao** manufacturer unknown | China 1963 | *Colletotrichum gloeosporioides* f. sp. *Cuscutae* | *Cuscuta chinensis* Lam., *Cuscuta australis* R. Br. | [30,183] |
| **DeVine** Valent Bioscences Crop | USA 1981 | strain MVW *Phytophthora palmivora*, *Phytophthora citrophthora* | *Morrenia odorata* Hook. &Arn. | [5,30,142,183] |
| **Collego (Lockdown)** Encore Technologies (Natural Industries) | USA 1982 (2006) | strain ATCC 20,358 *Colletotrichum gloeosporioides* f. sp. *aeschynomene* | *Aeschynomene virginica* L. | [5,30,142,183] |
| **Casst** manufacturer unknown | USA 1983 | *Alternaria cassiae* | *Cassia* spp. | [30,183] |
| **ABG-5003** manufacturer unknown | USA 1984 | *Cercosporarodmanii* | *Echhornia crassipes* Mart. Solms | [30,183] |
| **Velgo** manufacturer unknown | Canada 1987 | *Colletotrichum coccodes* | *Abutilon theophrasti* Medik. | [30,183] |
| **Dr.BioSedge** manufacturer unknown | USA 1987 | *Puccinia canaliculata* | *Cyperus esculentus* L. | [30,183] |
| **BioMal** Philom Bios (Novozymes) | Canada 1992 | strain ATCC 20,767 *Colletotrichum gloeosporioides* f. sp. *malvae* | *Malva pussila* Sm. | [5,30,142,183] |
| **Stumpout** PPRI Weed Pathology Unit, Stellenbosch | South Africa 1997 | *Cylindrobasidium laeve* | *Poa annua* L., *Acacia* sp. | [30,183] |
| **BioChon** manufacturer unknown | Netherlands Canada 1997 | *Chondrostereum purpureum* | *Prunus serotina* Ehrh. | [30,183] |
| **Hakatak** manufacturer unknown | South Africa 1999 | *Colletotrichum acutatum* | *Hakea gummosis*, *Hakea sericea* Schrad. and J.C.Wendl. | [30] |
| **MycoTech Paste** Mycoforestis Corp | Canada 2002 | strain HQ1 *Chondrostereum purpureum* | deciduous trees | [5,30,142,183] |
| **Woad Warrior** Greenville Farms | USA 2002 | *Puccinia thlaspeos* | *Isastis tinctoria* L. | [5,30,142,183] |
| **Chontrol (Ecoclear)** MycoLogic Inc. | Canada (USA) 2004 (2007) | strain PFC 2139 *Chondrostereum purpureum* | deciduous trees | [5,30,183] |
| **Smoulder** Loveland Products Inc. | USA 2005 | strain 059 *Alternaria destruens* | *Cuscuta* spp. | [5,30,142,183] |
| **Sarritor** Sarritor Inc. | Canada 2007 | strain IMI 344,141 *Sclerotinia minor* | *Taraxacum officinale* F.H.Wigg. | [5,30,142,183] |
| **Striga** manufacturer unknown | Africa 2008 | *Fusarium oxysporum* f sp. *stigae* | *Striga hermonthica* Delile Benth., *Striga asiatica* L. Kuntze | [30] |
| **Name unknown** The Scotts Company | Canada USA 2012 | strain 94-44B *Phoma macrostoma* | cotyledonous species | [5,30,183] |
| **Gibbatrianth** manufacturer unknown | India 2014 | *Gibbago trianthemae* | *Trianthema portulacastrum* L. | [30] |

*4.4. Problems with the Commercial Availability of Bioherbicides*

Despite continuous research, many bioherbicides have not been commercially approved at the global scale. Kumar et al. [30] distinguished four groups of limitations (biological, environmental, technological, and commercial) that inhibit the development of bioherbicides.

Biological limitations are related to their specificity, resistance mechanisms in segetal plants, and interactions with other organisms. Even under ideal conditions, most phytopathogenic organisms can only partially eliminate one weed species due to resistance variability and the presence of resistant biotypes in a particular group of weed species. Another problem is the possible exchange of genes between the isolates of the microorganisms that the bioherbicides are based on and the pathogenic strains that attack the crops [176].

According to Kumar et al. [30] and Abbas [184], climate-dependent environmental barriers that affect the stability and spread of phytopathogens also hinder the registration of bioherbicides for commercial use. It is a major challenge to provide microorganisms with ideal environmental conditions (optimal pH, temperature, and humidity) so that they can develop and exhibit effective herbicidal activity [177]. It is also important to ensure the correct balance of nutrients, including the carbon to nitrogen ratio. Moreover, it is difficult to design bioherbicides that eliminate aquatic weeds, because they must be able to cope with changing conditions both at the surface of the water and at the bottom of waterbody. These products are significantly influenced by temperature, salinity, oxygen concentration and light intensity [176].

The transfer from the laboratory to industrial production is a huge technological challenge in the development of bioherbicides [30]. Many preparations are highly effective under controlled conditions, but do not meet expectations under real field conditions. Flores-Vargas et al. [149] studied the influence of rhizospheric bacteria on the development of *R. raphanistrum*, *L. rigidum* and *Arctotheca calendula* L. Under laboratory conditions, 74 strains of tested rhizobacteria were observed to limit the growth of weed seedlings, yet under real field conditions, only 19 microbial isolates were able to influence the development of the shoots and roots of the weed plants. Kennedy et al. [185] showed that the *P. fluorescens* D7 strain limited the germination and root growth of plants in vitro more than in soil. In the laboratory, the analysed strain inhibited the growth of all the weed species tested, but in the plant–soil test, the D7 strain only significantly reduced the root growth of *Bromus* spp. The tested microorganisms did not significantly inhibit the development of *Triticum*, *Hordeum*, and *Aegilops cylindrica* host cultivars or *Agrostis gigantea* Roth. The authors suggested several different factors that could affect the loss of herbicidal activity of the rhizobacterial strains studied in the plant–soil test. First, microorganisms in the substrate and on plant roots compete with the introduced microorganisms for space, carbon, energy and nutrients. Second are colonisation indicators, such as soil properties and structure, as well as the quality and quantity of the root exudate. Third are compounds that inhibit the growth and development of plants, which can be produced and secreted by microorganisms only under certain environmental conditions. It was found that the speed of movement of microorganisms and the substances that they produce may be greater in vitro than in the soil. Reinhart et al. [186] also investigated the influence of *P. fluorescens* bacteria on the growth and germination of *B. tectorum*. According to their observations, strains ACK55 and D7 limited the germination and growth of plant organs only under laboratory conditions. The studies conducted in the natural environment showed that none of the previously mentioned bacterial strains inhibited the growth of shoots and roots of *B. tectorum*. The authors of this study did not specify a particular reason for the lack of herbicidal activity of rhizobacteria in the natural environment: possible factors that influenced this phenomenon could be the structural and chemical properties of the soil, as well as the competing microorganisms in the soil or the microorganisms that were introduced into the soil from the air or water.

Unpublished studies conducted at the Department of Soil Science and Microbiology on the influence of 40 bacterial and fungal isolates on the growth of weeds (*A. arvensis*,

*A. retroflexus*, *S. pumila*, and *A. myosuroides*) confirmed the above observations. Under laboratory conditions, 50% of the tested isolates inhibited the growth of the tested weed species (Figure 1), while their effectiveness was not confirmedin vivo.

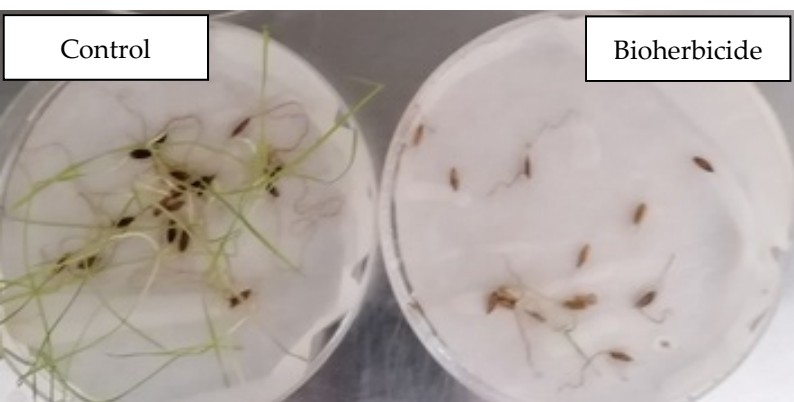

**Figure 1.** Inhibition of the germination of *A. myosuroides* Huds. by a bacterial bioherbicide (strain B1). Research at the Department of Soil Science and Microbiology at UP Poznań in 2021/2022 (own source).

According to Duke et al. [179], another technical challenge in the production of bio-preparations is to ensure the correct conditions for the multiplication of the microorganisms. According to the authors, the consequences of an improperly carried out process are the formation of an insufficient amount of mycelium and reduced spore viability, as well as the lack and limited number of natural phytotoxins produced by the microorganisms. It is also very important to develop the final product formula, which will ensure a stable and effective herbicide and the longest possible period of use and storage [184]. In practice, the application of bioherbicides to areas infested by weeds should be as straightforward as the application of chemical herbicides [30].

There are also commercial restrictions on the use of bioherbicides. These are mainly restrictions related to the crop protection product market and legal regulations. Each jurisdiction controls the development and commercialisation of biopreparations, as well as its own biopesticide registration procedures. This means that there is no universal law that regulates the production of bioherbicides and their sale on the world market. Microorganisms produce various by-products that could be harmful and toxic to the environment and human health. Therefore, it is important to undertake a risk analysis [177], which should be based on current experimentation, as well as on documentation from previous studies. However, this documentation is usually insufficient or absent. In the United States and Australia, the potential for research on the biocontrol of weeds is decreasing. The main reasons are the lack of documentation from previous experiments, frequent failures in the commercialisation of biopesticides, and time-consuming experiments to identify suitable microorganisms. A further problem is poor public awareness and lack of interest in biological pesticides from potential customers [30].

According to Duke et al. [179], many potential bioherbicides are not authorised for industrial production and some have been withdrawn from the global market due to the above-mentioned biological, environmental, technical and commercial barriers and the limited or poor efficacy of the bioproduct.

## 5. Conclusions

Extensive analyses of the toxicity of herbicides have led to a ban on their use in an increasing number of countries. Therefore, it is crucial to identify alternatives to traditional crop protection products, including herbicides. In response to this demand, researchers have conducted research on the production of biopreparations based on microorganisms and their metabolites, i.e., bioherbicides.

The non-toxicity of bioherbicides to the environment, humans and animals is a key aspect in favour of their implementation in weed control. However, despite continuous research, many have not been registered for commercial use, as it is difficult to verify whether the biological agents meet stability requirements. Fluctuations in temperature, pH or water content in the environment may cause changes in the activity and effectiveness of bioherbicides. It seems that the effectiveness of bioherbicides might be increased by combining them with adjuvants, i.e., auxiliary substances that strengthen and accelerate the effect of herbicidal preparations and prevent them from being washed off the plant. Nevertheless, the greatest challenge in the development of bioherbicides seems to be how to increase public awareness of the effectiveness and safety of these biopreparations.

**Author Contributions:** A.K. and A.W.-M. prepared and wrote the original draft of the manuscript; A.W.-M. and A.N. contributed to thereview and editing of the manuscript; A.K. and A.A.P. contributed to visualisation; A.W.-M., A.A.P. and A.N. contributed to funding acquisition; A.K. and A.W.-M.contributed to project administration. All authors have read and agreed to the published version of the manuscript.

**Funding:** This publication was co-financed within the framework of the Polish Ministry of Science and Higher Education's program: "Regional Excellence Initiative" in the years 2019–2022, project no. 005/RID/2018/19, financing amount 12,000,000.00 PLN.

**Conflicts of Interest:** The authors declare no conflict of interest.

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
