# Peer review of "The Problem of Weed Infestation of Agricultural Plantations vs. the Assumptions of the European Biodiversity Strategy"

_agronomy, doi:10.3390/agronomy12081808_

Round 1
Reviewer 1 Report
The manuscript is well written and organized. References are up-to-date and comprehensive. The theme is fundamental. Therefore, my recommendation is for the approval of the manuscript as it is.
Author Response
"Please see the attachment."

Reviewer 2 Report
The manuscript with the title „The Problem of Weed Infestation of Agricultural Plantations vs the Assumptions of the European Biodiversity Strategy” is a review article containing 214 references. My review will be short, because I found already in the first 20 citations many mistakes. I will not proceed to read and check in detail.
I miss the “mission” of the manuscript. There is no red line through. The manuscript is much to long. I have the impression – even though my English is not the best – that some of the criticised parts may result from translation errors. Therefore, an English proof-reading could help. On the other hand, the manuscript was good to read for me. I think some of the tables could be moved to the supplement. Maybe even large parts of the chapters 2 and 3.
Citations:
· in some cases you cite a reference that states something without giving a proof (i.e. L44, ref 8 is not about food security and wheat. It is about tank mixtures).
· In some cases you cite not precisely: in L. 76 you claim that the ref 13 is conduction experiments in central and eastern Poland. The ref states that they conducted ONE trial at ONE site.
· In some cases you extrapolate in an inaccurate way: in many citations in chapter 2.1 (“The most common weeds …”) you use the results of field trials to determine the “main weeds”. Theses trials are in most (all?) cases not intended to give an overfiew on the weed species in a region or country. They are conducted for other purposes. You can not claim to know the weed infestation of a region, if you have seed the results of a single trial at a single site. But there ARE approaches with this aim. Just to name some authors: G. Pinke, S. Lososova, G. Fried. But there are many more.
· Please check whether you need that many references.
· I stopped checking the references after chapter 2.1
1. Introduction
It is not clear what the authors aim at. The introduction needs to be clear on that point and can/must be extended quit a bit.
2. “Weed infestation of agricultural…”:
Again the aim is not clear with this very long chapter. What do you need the main weeds for anyway? Why do you lay such emphasis to define the main weeds?
I like the list of allelopathic weeds and their chemicals at L 361.
I do not see a reason to compare the two approaches to classify herbicides, at least to bring both graphs.
All together this chapter is to long and no direction is given. Please reduce this chapter to the necessary extend.
3. “Chemical herbicides”
All together this chapter is to long and no direction is given. Please reduce this chapter to the necessary extend. What for do you talk about the very old MOA in this detail?
4. This seems to be the core of the manuscript. You should make it much more clear.
5. Conclusion
The first paragraph in this chapter could be part of the introduction – I this is the aim with the current manuscript.
Author Response
"Please see the attachment."

Reviewer 3 Report
The paper is well documented and the subject is topical. Even if it is a review paper, the recommended structure for the Agronomy journal must be applied. The abstract must contain more specific information and concrete results, not just generalities.
It would be interesting to add the hypothesis from which the study started and there is no discussion section in which the authors can bring their own opinins following the bibliographic study.
Author Response
"Please see the attachment."
